# WINT3R: WINDOW-BASED STREAMING RECONSTRUCTION WITH CAMERA TOKEN POOL

**Zizun Li**[1,2]   **Jianjun Zhou**[2,3,4]   **Yifan Wang**[2,6]   **Haoyu Guo**[2]   **Wenzheng Chang**[2,6]
**Yang Zhou**[2,5]   **Haoyi Zhu**[1,2]   **Junyi Chen**[2,6]   **Chunhua Shen**[4]   **Tong He**[2,3,†]
[1]University of Science and Technology of China  [2]Shanghai AI Lab  [3]Shanghai Innovation Institute
[4]Zhejiang University  [5]Fudan University  [6]Shanghai Jiao Tong University  [†]Corresponding Author

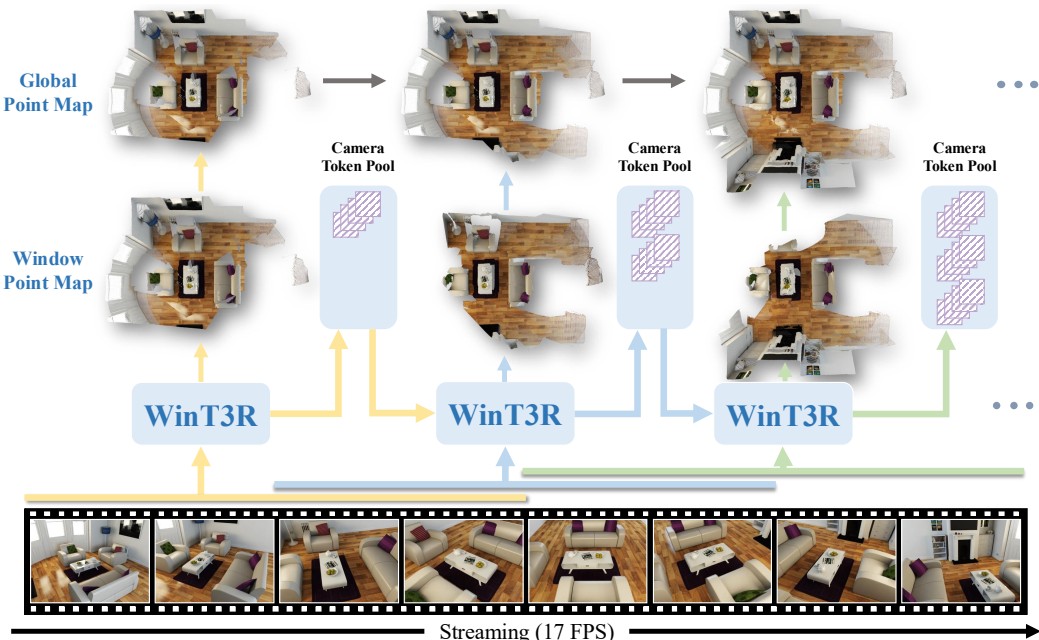

Figure 1: **Overview.** Given an image stream, our method WinT3R processes input images in a sliding-window manner, where adjacent windows overlap by half of the window size. Unlike previous online reconstruction methods, our model generates extremely compact camera tokens during online reconstruction to serve as global information for historical frames. This enables the reconstructions of subsequent windows to leverage these global cues for more accurate results. Our model achieves high-quality geometry reconstruction while maintaining real-time performance at 17 FPS.

## ABSTRACT

We present WinT3R, a feed-forward reconstruction model capable of online prediction of precise camera poses and high-quality point maps. Previous methods suffer from a trade-off between reconstruction quality and real-time performance. To address this, we first introduce a sliding window mechanism that ensures sufficient information exchange among frames within the window, thereby improving the quality of geometric predictions without introducing a large amount of extra computation. In addition, we leverage a compact representation of cameras and maintain a global camera token pool, which enhances the reliability of camera pose estimation without sacrificing efficiency. These designs enable WinT3R to achieve state-of-the-art performance in terms of online reconstruction quality, camera pose estimation, and reconstruction speed, as validated by extensive experiments on diverse datasets. Code and models are publicly available at https://github.com/LiZizun/WinT3R.

# 1 INTRODUCTION

Real-time reconstruction of 3D geometry from image streams is a fundamental problem with numerous practical applications. This task requires incrementally integrating newly arrived frames into existing reconstructions within a unified coordinate system at high speed. A typical approach involves traditional SLAM methods (Mur-Artal et al., 2015; Davison et al., 2007; Engel et al., 2014; Forster et al., 2016; Teed & Deng, 2021), which first extract features for tracking, then perform Bundle Adjustment (BA) to jointly refine camera poses and sparse 3D structures, and finally employ loop-closure detection to mitigate accumulated drift. While they achieve real-time localization and sparse mapping, they are not suitable for online dense reconstruction.

With the rapid advances in deep learning, some recent approaches demonstrate promising reconstruction capabilities, yet they face a trade-off between reconstruction quality and real-time performance. Specifically, offline methods (Wang et al., 2025a;d; Zhang et al., 2025; Yang et al., 2025) achieve high-quality reconstruction by performing full attention across image tokens of all frames. They fail to achieve real-time performance and cannot flexibly incorporate new frames into existing reconstruction results. In contrast, online methods (Liu et al., 2025; Wang & Agapito, 2024; Chen et al., 2025b; Wu et al., 2025; Zhuo et al., 2025; Team et al., 2025) like CUT3R (Wang et al., 2025b) achieve real-time reconstruction in a streaming manner by enabling image tokens from each new frame to interact with the state tokens. However, due to the lack of direct and sufficient interaction between image tokens of adjacent frames, the reconstruction quality remains suboptimal compared with offline methods.

To overcome these challenges, we propose WinT3R, a real-time and high-quality 3D reconstruction method based on a sliding-window strategy and a camera-token pool mechanism. Our design is motivated by two key observations. First, adjacent frames typically exhibit strong correlations, thus, the quality of geometric predictions can be improved if the image tokens can directly interact with those from neighboring frames. Second, camera tokens can be represented much more compactly than image tokens, which enables direct interaction with all historical frames without compromising real-time performance, thereby yielding more reliable camera pose estimation with a global perspective.

Based on these observations, we first propose an online sliding-window mechanism that processes input image streams in real time. Within this design, image tokens interact not only with the state tokens but also directly with other image tokens within the same window. Moreover, we maintain a compact camera token for each frame and store them in an expandable pool. When estimating the camera parameters for newly arrived frames, the model leverages all historical camera tokens in the pool, thus achieving more accurate estimates within real-time computational constraints.

We train our model using a variety of public datasets (Baruch et al., 2021; Dai et al., 2017; Li & Snavely, 2018; Li et al., 2023; Reizenstein et al., 2021; Roberts et al., 2021; Wang et al., 2020; Yeshwanth et al., 2023; Xia et al., 2024; Yao et al., 2020) and our private synthetic datasets. Experiments demonstrate that our model effectively mitigates the aforementioned issues and processes input image streams in real time at over 17 FPS while accurately predicting camera poses and point maps, thereby achieving state-of-the-art performance in online reconstruction tasks.

Our main contributions are summarized as follows:

1. We propose an online window mechanism, enabling sufficient interaction of image tokens within the same window and across adjacent windows.

2. We maintain a camera token pool, which functions as a lightweight "global memory" and improves the quality of camera pose prediction with a global perspective.

3. Experiments demonstrate that WinT3R achieves state-of-the-art performance in online 3D reconstruction and camera pose estimation, with the fastest reconstruction speed to date.

# 2 RELATED WORK

**Structure from Motion (SfM)** aims to jointly reconstruct 3D scene structures and camera poses from multi-view images (He et al., 2024; Zhang, 1997; Wang et al., 2024a; Agarwal et al., 2011). This task poses severe challenges due to the scale and complexity of real-world scenes. Traditional

approaches are categorized as incremental methods (Snavely, 2008; Schonberger & Frahm, 2016; Snavely et al., 2006; Wu et al., 2011), which progressively align images via iterative bundle adjustment (Hartley, 2003) but suffer from error accumulation; global methods (Govindu, 2004; Arie-Nachimson et al., 2012; Crandall et al., 2012), which directly optimizes global camera poses but remains sensitive to erroneous pairwise constraints; and hybrid methods (Cui et al., 2017; Moulon et al., 2013) that combine both paradigms to improve scalability. Recent advancements integrate deep learning to enhance robustness: Learned features (DeTone et al., 2018; Sun et al., 2021) and matchers (Sarlin et al., 2020; Lindenberger et al., 2023; Li et al., 2025) improve correspondence reliability, while differentiable optimization frameworks (Tang & Tan, 2018; Brachmann & Rother, 2021) enable end-to-end trainable pipelines. Despite progress, challenges remain in dynamic scenes, textureless regions, and the generalizability of learning-based methods beyond synthetic data.

**Multi-view Stereo** (MVS) methods (Furukawa & Ponce, 2009; Campbell et al., 2008) predominantly adopt a depth-map fusion paradigm, where depth maps are estimated per view and merged into a unified 3D reconstruction. Early approaches (Liu et al., 2009; Wang et al., 2021) iteratively propagate depth hypotheses via randomized initialization and cost aggregation. While efficient, these methods struggle with textureless regions and occlusions due to reliance on handcrafted similarity metrics. The advent of deep learning catalyzed significant advancements: MVSNet (Yao et al., 2018) pioneered cost-volume construction via differentiable homography warping and 3D CNN regularization, establishing an end-to-end trainable framework. Recently, direct RGB-to-3D methods like DUSt3R (Wang et al., 2024b) and MASt3R (Leroy et al., 2024) estimate point clouds from a pair of views, but they require additional global alignment process to handle multi-view tasks. Offline methods like VGGT (Wang et al., 2025a), FLARE (Zhang et al., 2025) and $\pi^3$ (Wang et al., 2025d) move a step forward DUSt3R (Wang et al., 2024b) to operate on multi-view images, but they cannot dynamically add new estimations to previous results.

**Online Reconstruction Methods** encompass simultaneous localization and mapping (SLAM) (Zhang & Singh, 2015; Shan et al., 2021; Engel et al., 2014; Zhu et al., 2022) and dynamic scene reconstruction (Yu et al., 2018; Bescos et al., 2018). Monocular SLAM systems estimate ego-motion and 3D structure in real time from video, but they generally assume known camera intrinsics. Recent learning-based methods (Civera et al., 2008; Tateno et al., 2017; Yang & Scherer, 2019; Team et al., 2025; Chen et al., 2025a) have bridged scalability and flexibility. MASt3R-SLAM (Murai et al., 2025) exploits a dense dual-view 3D reconstruction prior (building on DUSt3R (Wang et al., 2024b)/MASt3R (Leroy et al., 2024)) for real-time monocular SLAM. It models scenes with generic camera geometry, unifying pose estimation, dynamic point-cloud fusion, and loop closure. Innovations like CUT3R (Wang et al., 2025b) and Spann3R (Wang & Agapito, 2024) enabled feed-forward reconstruction from video sequences. Fully depending on memory or state tokens, these methods suffer from severe geometric distortions. In contrast, our compact representation of camera tokens and local point maps alleviates this problem, yielding superior reconstruction quality.

## 3 METHOD

Given a stream of input images, WinT3R predicts local point map and camera pose for each frame in real-time, as illustrated in Figure 2. We first propose an online window mechanism to process images in a sliding window manner, facilitating information exchange within the window and enriching image tokens with state tokens (Section 3.1). Next, we predict the local point map for each frame through a lightweight convolutional head and estimate the camera pose for each frame based on a camera token pool (Section 3.2). Finally, we describe our training objectives (Section 3.3).

### 3.1 ONLINE WINDOW MECHANISM

The input is a stream of $(\boldsymbol{I}_i)_{i=1}^{T}$ of RGB images $\boldsymbol{I}_i \in \mathbb{R}^{3 \times H \times W}$, observing the 3D scene. For each coming image $\boldsymbol{I}_i$, we first send it to a ViT encoder to obtain the image token $\boldsymbol{F}_i \in \mathbb{R}^{N \times C}$:

$$\boldsymbol{F}_i = \text{Encoder}(\boldsymbol{I}_i). \tag{1}$$

Inspired by CUT3R (Wang et al., 2025b), we maintain a set of state tokens $\boldsymbol{S}$ for the scene, which allow image tokens to read contextual information and simultaneously update these state tokens. However, in CUT3R, information between frames can only be shared indirectly through these state tokens. To leverage the strong correlation among adjacent frames, we introduce a sliding window

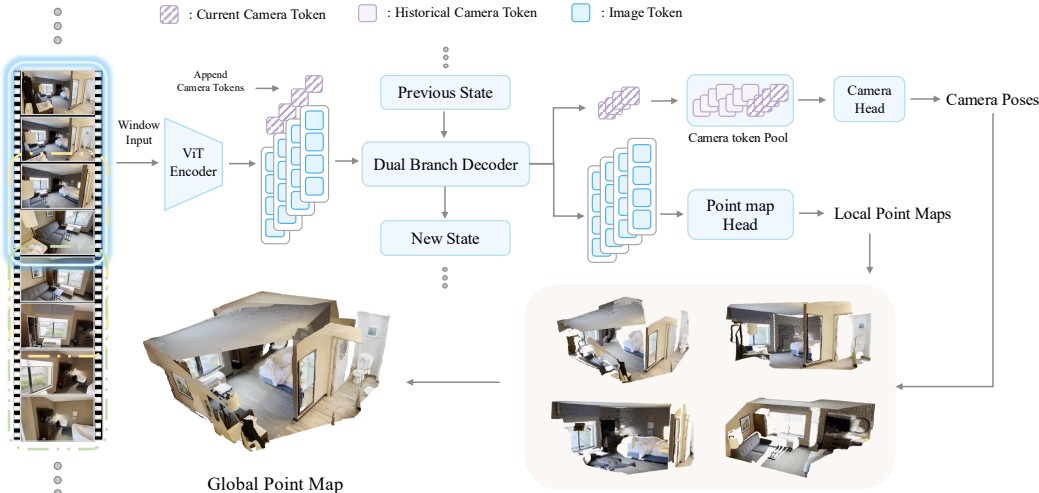

Figure 2: **WinT3R pipeline**. We detail the reconstruction process within a single window. All images are first passed through a frame-wise ViT encoder, which outputs image tokens. Camera tokens are then appended to these tokens. Then the tokens within this window are collectively fed into a decoder to interact with state tokens. Finally, the image tokens output by the decoder are sent to a lightweight convolutional head to predict local point maps. Meanwhile, the camera tokens, along with those in the camera token pool, are jointly fed into a camera head to predict camera parameters, while these camera tokens are simultaneously added to the camera token pool.

mechanism to facilitate more direct cross-frame communication between image tokens and state tokens, thereby enhancing prediction quality. Specifically, for the input image stream, we set a sliding window of size $w$. During each interaction step, to enable comprehensive information exchange across frames, all image tokens in the current window are used as input.

$$[\boldsymbol{g}_i^g, \boldsymbol{F}_i^g]_{i \in \mathcal{W}_t}, [\boldsymbol{g}_i^l, \boldsymbol{F}_i^l]_{i \in \mathcal{W}_t}, \boldsymbol{S}_t = \text{Decoders}([\boldsymbol{g}_i, \boldsymbol{F}_i]_{i \in \mathcal{W}_t}, \boldsymbol{S}_{t-1}), \tag{2}$$

where $\mathcal{W}_t$ denotes the current window, and $\boldsymbol{g}_i$ denotes the learnable camera token prepended to the image tokens $\boldsymbol{F}_i$, which is used for camera pose prediction. The decoder is equipped with two branches interconnected with each other. One branch inputs image tokens and camera tokens, which is designed to perform Alternating-Attention as VGGT (Wang et al., 2025a) and outputs both global ($\boldsymbol{g}_i^g$ and $\boldsymbol{F}_i^g$) and local ($\boldsymbol{g}_i^l$ and $\boldsymbol{F}_i^l$) enriched tokens for these frames. The other branch inputs state tokens $\boldsymbol{S}_{t-1}$ and outputs updated tokens $\boldsymbol{S}_t$ which have exchanged information with the image tokens within the window $\mathcal{W}_t$. Specifically, we initialize the state tokens as a set of learnable tokens at the beginning of the reconstruction process.

With this design, the image tokens can not only read contextual information from the state tokens, but also interact directly with other tokens in the current window. Furthermore, to enhance continuity between adjacent windows, we set the sliding window stride to $w/2$, ensuring neighboring windows share half of their frames. This design allows predictions for the overlapping region to be updated based on subsequent $w/2$ frames.

To balance the real-time requirements of online processing and the reconstruction performance of the model, we select a window size of 4 and a stride of 2 in our implementation. During the inference process, we check if the window is full. If not, current image tokens will wait for subsequent images to arrive until the window reaches the full size. For the last image, we duplicate it to fill the remaining window slots. Regarding the overlapping region between the initial prediction and the updated prediction, we select the camera pose from the updated prediction and the point map with the higher confidence score as the final output.

### 3.2 POINT MAP AND CAMERA PREDICTION

Based on the enriched image and camera tokens, we predict the point map $\hat{\boldsymbol{P}}_i$ and camera pose $\hat{c}_i$ for each frame. The point map of each frame is defined in its own local camera coordinate system, which

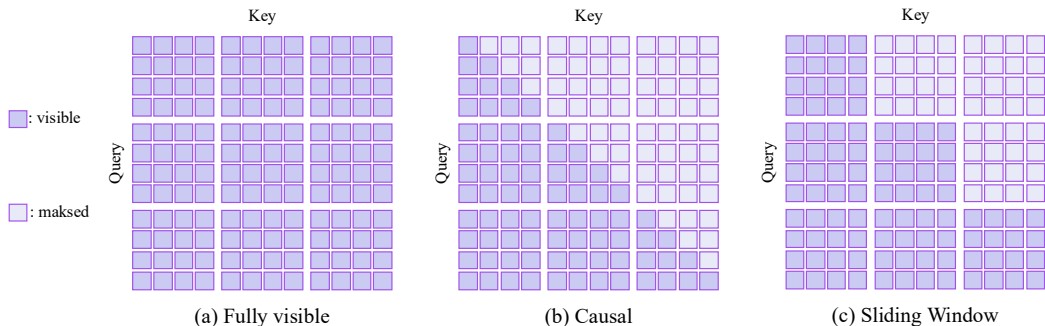

Figure 3: **Attention mask.** (a) Full attention, all input tokens are covisible. (b) Causal attention, each token can only see itself and the tokens before it in the sequence. (c) Sliding window attention, each token can only see tokens in current window and the tokens in history windows.

mainly contains local geometric information, so we consider the prediction relies primarily on local cues. Since the image tokens $F_i^l$ have already captured sufficient contextual information through interactions with the state tokens $S_{t-1}$ and other image tokens within the window, we directly feed them into the point map head to predict the local point map $\hat{P}_i$ and its corresponding confidence $C_i$. To optimize efficiency and quality, we avoid the computationally expensive DPT head and the linear head which introduces grid-like artifacts, opting instead for a lightweight convolutional head:

$$\hat{P}_i, C_i = \text{ConvHead}(F_i^l). \tag{3}$$

In contrast, the camera pose represents the position and orientation of each frame within the entire 3D scene. Therefore, predicting the camera pose requires a more comprehensive utilization of global information to achieve reliable results. To this end, we store all historical camera tokens in a pool and leverage all of them when predicting the camera pose for each incoming frame. Furthermore, to make camera tokens more expressive, we concatenate the local camera token $g_i^l$ and the global camera token $g_i^g$ along the channel dimension to form the final camera token $g_i'$.

$$g_i' = \text{ChannelCat}(g_i^l, g_i^g), \tag{4}$$

$$\text{Pool}_{cam}^t = \text{Pool}_{cam}^{t-1} \sqcup [g_i']_{i \in \mathcal{W}_t}, \tag{5}$$

$$[\hat{c}_i]_{i \in \mathcal{W}_t} = \text{CameraHead}([g_i']_{i \in \mathcal{W}_t}, \text{Pool}_{cam}^{t-1}). \tag{6}$$

Here the camera parameters $\hat{c}_i \in \mathbb{R}^7$ is the concatenation of rotation quaternion $q \in \mathbb{R}^4$ and translation $t \in \mathbb{R}^3$. $\sqcup$ indicates adding new calculated camera tokens to the pool.

For each frame, our model outputs only a single camera token $g_i'$, which is a 1536-dimensional vector in our implementation. The number of such camera tokens is significantly fewer than the number of image tokens, ensuring the real-time performance of our system. Considering that the output of the camera parameter $\hat{c}_i$ is only a 7-dimensional vector, which is of significantly lower-dimensional than the point map $\hat{P}_i \in \mathbb{R}^{3 \times H \times W}$, this compact token design does not compromise prediction accuracy. Compared with other methods like caching memory tokens that require storing all keys and values for every attention layer, our approach drastically reduces storage overhead and computational cost.

To better leverage these compact camera tokens, we design a camera head with sliding window masked attention that matches the decoder's architecture. Our attention mask is illustrated in Figure 3 (c). This attention mask enables the model to predict camera tokens of current window condition on all previous windows, without being affected by subsequent windows at training stage.

### 3.3 TRAINING OBJECTIVE

We train our model end-to-end using camera pose loss and point map loss:

$$\mathcal{L}_{\text{total}} = \mathcal{L}_{\text{camera}} + \mathcal{L}_{\text{pmap}}. \tag{7}$$

We normalize the prediction and ground truth respectively. Specifically, we first calculate the norm factors as the averaged point map scale weighted by confidence:

$$norm([\boldsymbol{P}_i]_{i=1}^T, [\boldsymbol{C}_i]_{i=1}^T) = \frac{\sum_{i=1}^T \sum_{j \in M_i} \boldsymbol{P}_{i,j} \log \boldsymbol{C}_{i,j}}{\sum_{i=1}^T \sum_{j \in M_i} \log \boldsymbol{C}_{i,j}}. \tag{8}$$

Then we normalize both the predicted and the ground-truth camera translations and point maps using the norm factors. The local point map loss includes a confidence-aware regression term as MASt3R (Murai et al., 2025):

$$\mathcal{L}_{\text{pmap}} = \sum_{i=1}^T \sum_{j \in M_i} \boldsymbol{C}_{i,j} \ell_{\text{regr}}^{\text{pmap}}(j, i) - \alpha \log \boldsymbol{C}_{i,j}, \tag{9}$$

where $M_i$ denotes the valid pixel mask. We apply $\ell_2$ loss for the point map regression term $\ell_{\text{regr}}^{\text{pmap}}$. Following $\pi^3$ (Wang et al., 2025d), we supervise the relative camera pose, avoiding manually defining a coordinate system. The network adaptively predicts camera poses in a learned coordinate frame. Consequently, we employ a relative camera pose loss, supervising the pairwise relative poses for all frames rather than the absolute pose of each frame. The pairwise relative camera parameters $\boldsymbol{c}_{ij}$ from view $i$ to $j$ for the predicted and the ground truth are the concatenation of relative rotation quaternion $\boldsymbol{q}_{ij} \in \mathbb{R}^4$ and relative translation $\boldsymbol{t}_{ij} \in \mathbb{R}^3$.

$$\boldsymbol{q}_{ij} = \boldsymbol{q}_j^* \otimes \boldsymbol{q}_i, \tag{10}$$

$$\boldsymbol{t}_{ij} = \text{rotate}(\boldsymbol{t}_i - \boldsymbol{t}_j, \boldsymbol{q}_j^*), \tag{11}$$

where $\boldsymbol{q}_j^*$ is the conjugate of $\boldsymbol{q}_j$ and $\otimes$ denotes quaternion multiplication, $\text{rotate}(\boldsymbol{t}, \boldsymbol{q})$ applies the rotation represented by quaternion $\boldsymbol{q}$ to translation $\boldsymbol{t}$. Our camera pose loss compares the predicted relative camera parameters $\hat{\boldsymbol{c}}_{ij}$ with the ground truth $\boldsymbol{c}_{ij}$ using $\ell_1$ Loss:

$$\mathcal{L}_{\text{camera}} = \frac{1}{N(N-1)} \sum_{i \neq j} \ell_1(\hat{\boldsymbol{c}}_{ij}, \boldsymbol{c}_{ij}). \tag{12}$$

In our implementation, we found that the supervision from both the $\ell_1$ based camera loss and point map loss is equally critical, so we simply add them to form the final loss.

## 4 EXPERIMENTS

### 4.1 TRAINING DATASETS

We train our model using a large collection of datasets, including: GTASfm (Wang & Shen, 2020), WildRGBD (Xia et al., 2024), CO3Dv2 (Reizenstein et al., 2021), ARKitScenes (Baruch et al., 2021), TartanAir (Wang et al., 2020), Scannet (Dai et al., 2017), Scannet++ (Yeshwanth et al., 2023), BlendedMVG (Yao et al., 2020), MatrixCity (Li et al., 2023), Taskonomy (Zamir et al., 2018), MegaDepth (Li & Snavely, 2018), Hypersim (Roberts et al., 2021), and a synthetic dataset of video games. Our datasets cover a wide range of scenarios, such as object level and scene level, real-world data and synthetic data, video sequences and multiview images. We employ three sampling strategies: random sampling, interval sampling, and overlap view sampling.

### 4.2 IMPLEMENTATION DETAILS

Our model is initialized with pretrained weights of DUSt3R (Wang et al., 2024b) and trained using AdamW (Loshchilov & Hutter, 2019) optimizer. The full model has 750 million parameters. We train our model in two stages. In the first stage, we train the model with 12-frame data for 100 epochs, setting the maximum learning rate to 1e-4 and using a batch size of 4 per GPU. This stage is conducted on 64 NVIDIA A800 GPUs and takes 7 days. In the second stage, we fine-tune the model using 60-frame data for 12 epochs, with a maximum learning rate of 2e-6, completing in 4 days on 32 A800 GPUs. All input images during training have variable aspect ratios, with the longest edge fixed at 512 pixels.

Table 1: **Quantitative 3D reconstruction results on DTU and ETH3D datasets.**

| Method | Type | DTU | | | ETH3D | | |
|---|---|---|---|---|---|---|---|
| | | Acc↓ | Comp↓ | Overall↓ | Acc↓ | Comp↓ | Overall↓ |
| Fast3R (Yang et al., 2025) | Offline | 3.083 | 2.329 | 2.706 | 0.638 | 0.738 | 0.688 |
| FLARE (Zhang et al., 2025) | Offline | 2.077 | 1.982 | 2.030 | 0.522 | 0.542 | 0.530 |
| VGGT (Wang et al., 2025a) | Offline | **1.140** | **1.439** | **1.289** | **0.186** | **0.144** | **0.165** |
| Spann3R (Wang & Agapito, 2024) | Online | 6.021 | 3.554 | 4.788 | 0.733 | 1.546 | 1.139 |
| SLAM3R (Liu et al., 2025) | Online | 6.672 | 5.256 | 5.964 | 0.626 | 0.888 | 0.757 |
| CUT3R (Wang et al., 2025b) | Online | 4.454 | 1.944 | 3.199 | 0.533 | 0.503 | 0.518 |
| Point3R (Wu et al., 2025) | Online | 4.887 | 1.688 | 3.288 | 0.662 | 0.579 | 0.621 |
| StreamVGGT (Zhuo et al., 2025) | Offline | 3.997 | **1.651** | 2.823 | 0.581 | 0.359 | 0.470 |
| **Ours** | Online | **3.638** | 1.838 | **2.738** | **0.411** | **0.272** | **0.341** |

Table 2: **Quantitative 3D reconstruction results on 7-Scenes and NRGBD datasets.**

| Method | Type | 7-Scenes | | | NRGBD | | |
|---|---|---|---|---|---|---|---|
| | | Acc↓ | Comp↓ | Overall↓ | Acc↓ | Comp↓ | Overall↓ |
| Fast3R (Yang et al., 2025) | Offline | 0.040 | 0.059 | 0.049 | 0.074 | 0.052 | 0.063 |
| FLARE (Zhang et al., 2025) | Offline | **0.019** | 0.026 | **0.022** | 0.022 | 0.018 | 0.020 |
| VGGT (Wang et al., 2025a) | Offline | 0.023 | **0.026** | 0.025 | **0.017** | **0.015** | 0.165 |
| Spann3R (Wang & Agapito, 2024) | Online | 0.054 | 0.044 | 0.049 | 0.134 | 0.078 | 0.106 |
| SLAM3R (Liu et al., 2025) | Online | 0.069 | 0.060 | 0.064 | 0.130 | 0.082 | 0.106 |
| CUT3R (Wang et al., 2025b) | Online | 0.023 | 0.027 | 0.025 | 0.086 | 0.048 | 0.067 |
| Point3R (Wu et al., 2025) | Online | 0.034 | 0.026 | 0.030 | 0.066 | 0.032 | 0.049 |
| StreamVGGT (Zhuo et al., 2025) | Online | 0.047 | 0.030 | 0.038 | 0.096 | 0.049 | 0.074 |
| **Ours** | Online | **0.023** | **0.022** | **0.022** | **0.032** | **0.020** | **0.026** |

Table 3: **Camera Pose Estimation on Tanks and Temples, CO3Dv2 and 7-Scenes datasets.**

| Method | Type | Tanks and Temples | | | CO3Dv2 | | | 7-Scenes | | |
|---|---|---|---|---|---|---|---|---|---|---|
| | | RRA@30↑ | RTA@30↑ | AUC@30↑ | RRA@30↑ | RTA@30↑ | AUC@30↑ | RRA@30↑ | RTA@30↑ | AUC@30↑ |
| Fast3R (Yang et al., 2025) | Offline | 66.15 | 71.69 | 50.18 | 97.49 | 90.97 | 73.59 | 90.66 | 82.18 | 60.92 |
| FLARE (Zhang et al., 2025) | Offline | 85.37 | 87.62 | 70.97 | 96.35 | 93.52 | 73.79 | **100.0** | 95.68 | 75.90 |
| VGGT (Wang et al., 2025a) | Offline | **93.83** | **95.72** | **91.17** | **98.98** | **97.07** | **89.89** | **100.0** | **97.36** | **79.71** |
| Spann3R (Wang & Agapito, 2024) | Online | 65.52 | 68.54 | 40.78 | 93.81 | 89.95 | 70.41 | 99.98 | 95.10 | 72.60 |
| CUT3R (Wang et al., 2025b) | Online | 92.35 | 91.86 | 76.22 | 96.33 | 92.67 | 75.94 | **100.0** | 95.36 | 74.49 |
| Point3R (Wu et al., 2025) | Online | 74.64 | 79.27 | 42.63 | 95.51 | 91.21 | 67.99 | **100.0** | 94.13 | 66.81 |
| StreamVGGT (Zhuo et al., 2025) | Online | 93.23 | 92.81 | 74.98 | 98.61 | 95.60 | **84.68** | 99.98 | 95.78 | 75.50 |
| **Ours** | Online | **94.53** | **94.35** | **81.34** | **98.66** | **95.90** | 84.61 | **100.0** | **97.40** | **78.59** |

## 4.3 3D RECONSTRUCTION

Following the evaluation protocol of VGGT (Wang et al., 2025a), we evaluate 3D reconstruction quality on object-centric DTU (Jensen et al., 2014) and scene level ETH3D (Schops et al., 2017) datasets, reporting Accuracy, Completeness, and Overall (Chamfer distance) for point map estimation as VGGT. We sample keyframes every 2 images and align the predicted point maps and the ground truth using the Umeyama (Umeyama, 2002) algorithm. We further evaluate our method on scene-level 7-Scenes (Shotton et al., 2013) and NRGBD (Azinović et al., 2022) datasets, with a stride of 40 (7-Scenes) or 100 (NRGBD). We compare our method with other online reconstruction methods and offline reconstruction methods, as shown in Table 1, 2 and Figure 4, 5, our method demonstrates state-of-the-art performance among online methods across a broad spectrum of 3D reconstruction tasks, encompassing both real-world and synthetic data, at both object-level and scene-level.

## 4.4 CAMERA POSE ESTIMATION

For the camera pose estimation task, to ensure fair comparisons, we selected Tanks and Temples (Knapitsch et al., 2017), CO3Dv2 (Reizenstein et al., 2021), and 7-Scenes (Shotton et al., 2013) datasets for evaluation. All evaluated models have either been trained on these datasets or not at all. These datasets encompass both object-level and scene-level contexts, as well as real-world and synthetic data. For Tanks and Temples, we select 30 frames per scene with a stride of 10; for CO3Dv2, we randomly sample 10 frames per scene; for 7-Scenes, we sample frames with a

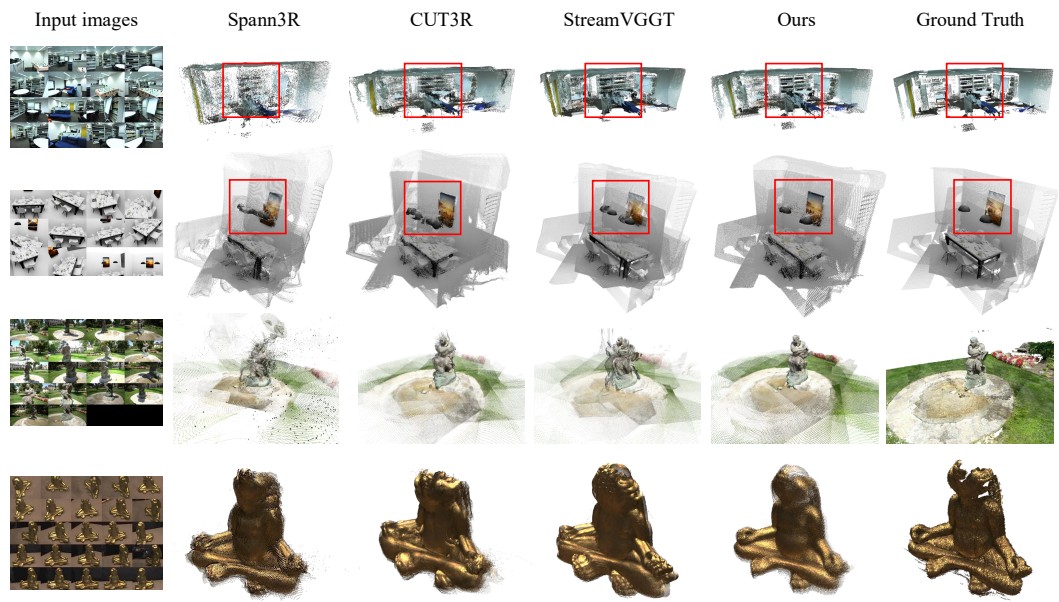

Figure 4: **Qualitative comparison of 3D reconstruction.** Compared with other online methods, WinT3R achieves higher reconstruction accuracy while also enabling faster reconstruction speed.

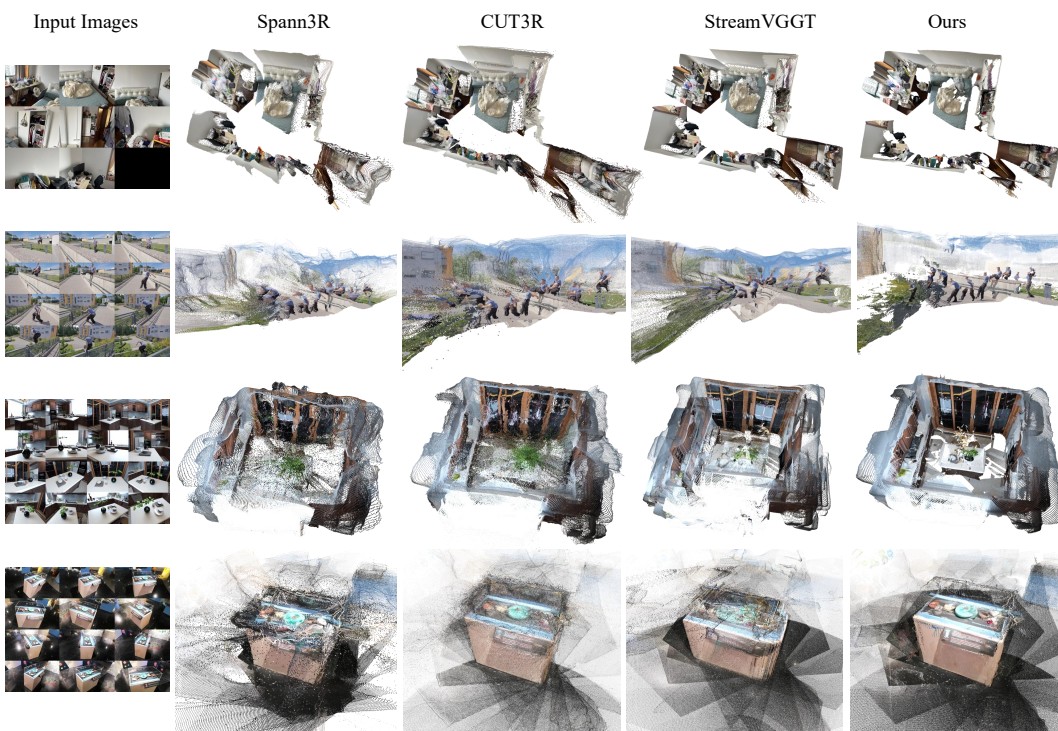

Figure 5: **Qualitative comparison of in-the-wild multi-view 3D reconstruction.** We demonstrate reconstruction results on in-the-wild sequences across indoor, outdoor, and object-level scenes. Our method consistently achieves the most photorealistic reconstruction results.

stride of 40. We evaluate them using Relative Rotation Accuracy (RRA) and Relative Translation Accuracy (RTA) at a given threshold (e.g., RRA@30 for 30 degrees), and AUC@30 which serves as a unified evaluation metric, defined as the area under the accuracy-threshold curve for the minimum of RRA and RTA across varying thresholds. The results in Table 3 show that our model delivers state-of-the-art performance among online methods.

Table 4: **Video Depth Estimation on Sintel, BONN and KITTI datasets.**

| Method | Type | Sintel | | BONN | | KITTI | | |
|---|---|---|---|---|---|---|---|---|
| | | Abs Rel↓ | $\delta < 1.25$↑ | Abs Rel↓ | $\delta < 1.25$↑ | Abs Rel↓ | $\delta < 1.25$↑ | FPS↑ |
| Fast3R (Yang et al., 2025) | Offline | 0.641 | 0.423 | 0.193 | 0.774 | 0.136 | 0.834 | 30.6 |
| FLARE (Zhang et al., 2025) | Offline | 0.729 | 0.336 | 0.152 | 0.790 | 0.356 | 0.570 | 3.9 |
| VGGT (Wang et al., 2025a) | Offline | **0.292** | **0.652** | **0.055** | **0.971** | **0.073** | **0.963** | **46.6** |
| Spann3R (Wang & Agapito, 2024) | Online | 0.597 | 0.384 | 0.072 | 0.953 | 0.251 | 0.566 | 10.4 |
| CUT3R (Wang et al., 2025b) | Online | 0.417 | 0.507 | 0.078 | 0.937 | 0.122 | 0.876 | 12.9 |
| Point3R (Wu et al., 2025) | Online | 0.461 | 0.455 | 0.060 | 0.962 | 0.137 | 0.839 | 3.6 |
| StreamVGGT (Zhuo et al., 2025) | Online | **0.343** | **0.604** | **0.057** | **0.974** | 0.185 | 0.700 | 13.7 |
| **Ours** | Online | 0.374 | 0.506 | 0.070 | 0.912 | **0.081** | **0.949** | **17.2** |

Table 5: **Ablation Study on 7-Scenes and NRGBD datasets.**

| Method | 7-Scenes | | | NRGBD | | |
|---|---|---|---|---|---|---|
| | Acc↓ | Comp↓ | Overall↓ | Acc↓ | Comp↓ | Overall↓ |
| $w/o$ pool | 0.126 | **0.200** | 0.163 | 0.220 | 0.480 | 0.350 |
| $w/o$ window | 0.123 | 0.300 | 0.212 | 0.253 | 0.556 | 0.404 |
| $w/o$ overlap | 0.126 | 0.265 | 0.195 | 0.220 | 0.349 | 0.285 |
| **Full model** | **0.118** | 0.205 | **0.161** | **0.217** | **0.298** | **0.258** |

Table 6: **Camera Pose Ablation on Tanks and Temples, CO3Dv2 and 7-Scenes datasets.**

| Method | Tanks and Temples | | | CO3Dv2 | | | 7-Scenes | | |
|---|---|---|---|---|---|---|---|---|---|
| | RRA@30↑ | RTA@30↑ | AUC@30↑ | RRA@30↑ | RTA@30↑ | AUC@30↑ | RRA@30↑ | RTA@30↑ | AUC@30↑ |
| $w/o$ pool | 28.24 | 40.93 | 8.87 | 76.01 | 78.23 | 38.10 | 65.38 | 41.22 | 11.54 |
| $w/o$ window | 30.69 | 43.77 | 12.05 | 74.54 | 75.63 | 37.83 | 47.76 | 32.69 | 7.39 |
| $w/o$ overlap | 30.13 | 44.83 | 11.83 | 81.23 | 80.44 | 44.31 | 56.34 | 40.98 | 11.54 |
| **Full model** | **35.88** | **51.32** | **15.73** | **83.54** | **81.98** | **47.17** | **67.92** | **43.32** | **15.01** |

## 4.5 VIDEO DEPTH ESTIMATION

We evaluate video depth estimation by aligning the predicted depth maps to the ground truth with a per-sequence scale. This alignment enables the assessment of both per-frame depth accuracy and inter-frame depth consistency. We report the Absolute Relative Error (Abs Rel) and the prediction accuracy in Table 4, the results show that our method demonstrates comparable or better performance than other online approaches. Furthermore, we also evaluate inference efficiency of KITTI (Geiger et al., 2013) dataset on a single NVIDIA A800 GPU, the result shows that our model runs at the highest speed among online reconstruction methods, running at 17.2 FPS.

## 4.6 ABLATION STUDIES

To quantify the contribution of each individual component, we conduct a series of ablation studies on our proposed method. Specifically, we remove each element in our model to validate the effectiveness of our designs. "$w/o$ pool" indicates that the camera head only uses the camera token within the current window for prediction, rather than conditions on camera tokens of all historical windows. "$w/o$ window" indicates the model inputs images frame by frame. "$w/o$ overlap" indicates that there is no overlapping between the frames of adjacent windows, the stride is set equal to the window size. In our ablation studies, all models were trained on $224 \times 224$ resolution from scratch without using any pretrained weights. For "$w/o$ pool", "$w/o$ overlap" and our full model, we set a window size of 4.

We first validate the effectiveness of our design in reconstruction quality on 7-Scenes and NRGBD datasets. To further verify the efficacy of our camera pose prediction design, we compare the pose estimation accuracy across all ablated models. As demonstrated in Table 5 and Table 6, the use of a camera token pool leads to a significant improvement in camera pose prediction accuracy. Our online window and online mechanism also significantly enhance the quality of 3D reconstruction.

## 5 CONCLUSION

In this paper, we propose WinT3R, an online model for continuous prediction of camera poses and point maps from streaming images. Our framework not only employs state tokens to align new reconstructions with existing scene geometry, but also utilizes camera tokens to compactly represent global information for each frame. This representation enables the model to capture global information of historical frames, drastically reducing storage overhead and computational costs. Furthermore, our overlapping sliding window strategy enhances continuity across consecutive windows, facilitating comprehensive information exchange. Experimental results demonstrate improvements in reconstruction accuracy and efficiency, validating the efficacy of our design for online 3D reconstruction tasks.

ACKNOWLEDGMENTS

This work is supported by Shanghai Artificial Intelligence Laboratory.

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

# A APPENDIX

## A.1 ARCHITECTURE DETAILS

The ViT encoder and state decoder maintain the same architecture as CUT3R (Wang et al., 2025b). The encoder has 1024 dimensions and 24 blocks, while the state decoder has 768 dimensions and 12 blocks. The image decoder employs an alternating attention mechanism with 768 dimensions and 12 blocks. The learnable state tokens are configured as 1024 tokens, each with 768 dimensions. For camera pose prediction, our prediction head is adapted from VGGT (Wang et al., 2025a), consisting of a 1536-dimensional, 4-layer transformer block, followed by an MLP layer to output the final camera parameters. Our point map head is a lightweight convolutional head adapted from MoGe (Wang et al., 2025c), modified to accept 768-dimensional input.

## A.2 LONG SEQUENCE COMPARISONS

In online settings, long sequence prediction is also highly important, and we have conducted additional comparisons for long sequences. We selected two models with relatively strong performance, CUT3R (Wang et al., 2025b) and StreamVGGT (Zhuo et al., 2025), for comparison. 8 compares the efficiency of the models when processing different numbers of frames at a resolution of 512×288.

Table 7: **Long Sequence Comparison on 7-Scenes and NRGBD datasets.**

| Method | 7-Scenes | | | NRGBD | | |
|---|---|---|---|---|---|---|
| | Acc↓ | Comp↓ | Overall↓ | Acc↓ | Comp↓ | Overall↓ |
| CUT3R (Wang et al., 2025b) | 0.083 | 0.042 | 0.062 | 0.194 | 0.089 | 0.142 |
| StreamVGGT (Zhuo et al., 2025) | 0.041 | **0.020** | **0.031** | 0.110 | **0.027** | **0.068** |
| **Ours** | **0.037** | 0.031 | 0.034 | **0.095** | 0.075 | 0.085 |

7 compares the quality of different models when processing 200 frames of data. The results show that our model maintains real-time performance even when predicting over hundreds of frames, while significantly outperforming CUT3R in prediction quality. Moreover, it achieves reconstruction quality close to that of StreamVGGT while being 14 times faster. The experimental results further demonstrate the compactness and effectiveness of our model's camera token pool.

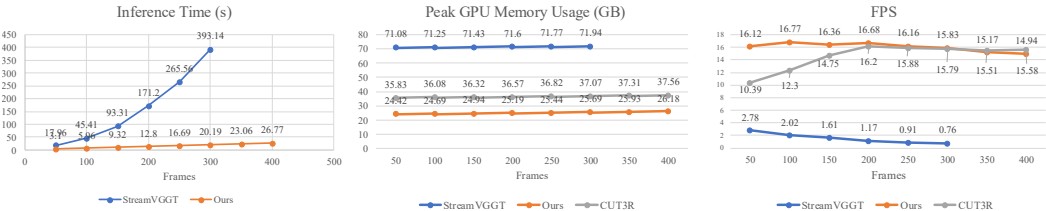

Figure 6: **Inference efficiency of 3D reconstruction.** We demonstrate inference efficiency of different frame numbers. Our method achieves almost the fastest and the most GPU memory efficient.

### A.3 LIMITATIONS

Our model, which utilizes compact camera tokens to assist in predicting camera parameters for subsequent frames, demonstrates promising results. However, it still struggles to avoid the issue of accumulated errors when processing very long videos or a large number of images. Secondly, during the training process, temporal data must be passed through the model sequentially, which requires longer training times compared to offline models. Designing a streaming model that conserves training resources remains a challenge to be addressed.

### A.4 LLM USAGE STATEMENT

In the writing of this paper, the LLM serves as a writing assistant, used for language translation and to provide concise, accurate, and academic language expression, as well as to correct grammatical errors. All the core ideas, experiments, formulas, methodologies, and figures in this paper originate from the authors and are independent of the LLM.

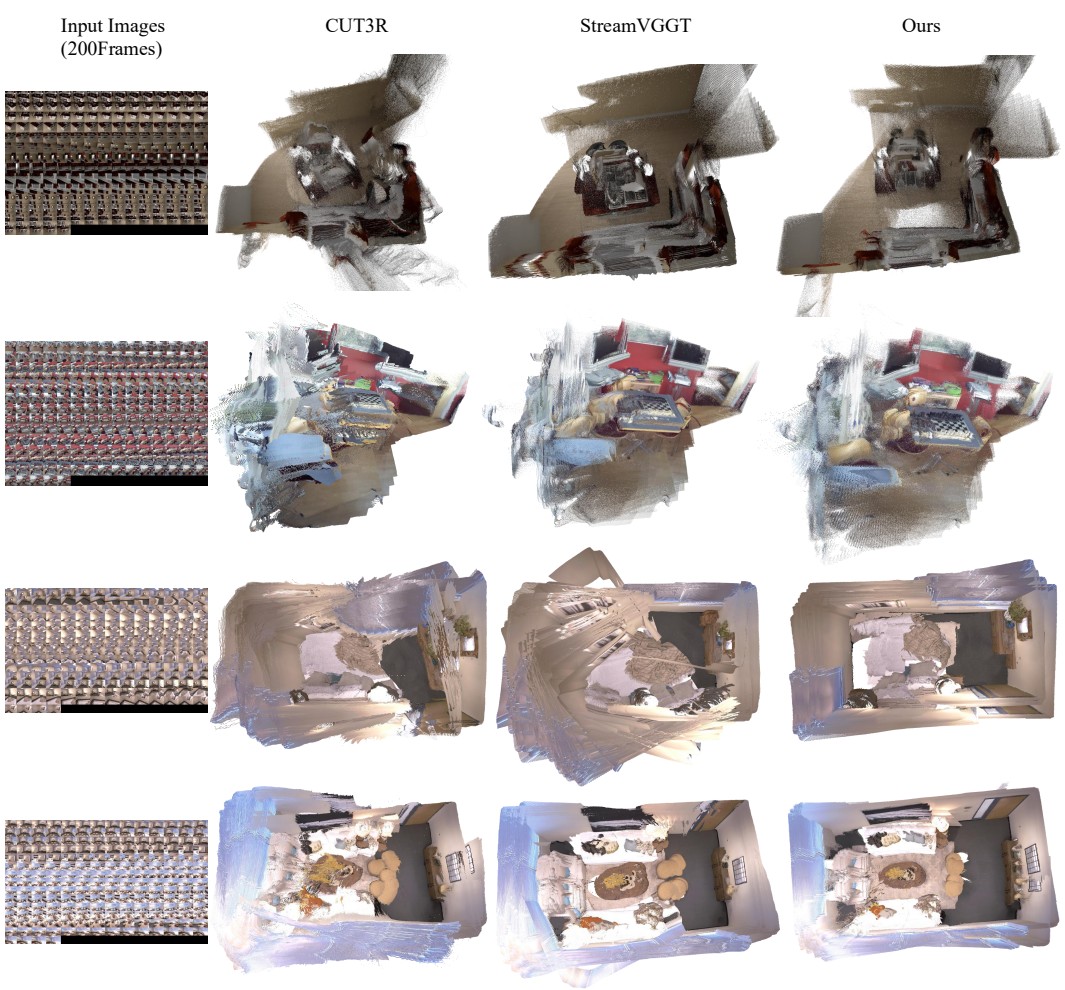

Figure 7: **Long sequence visualization.**

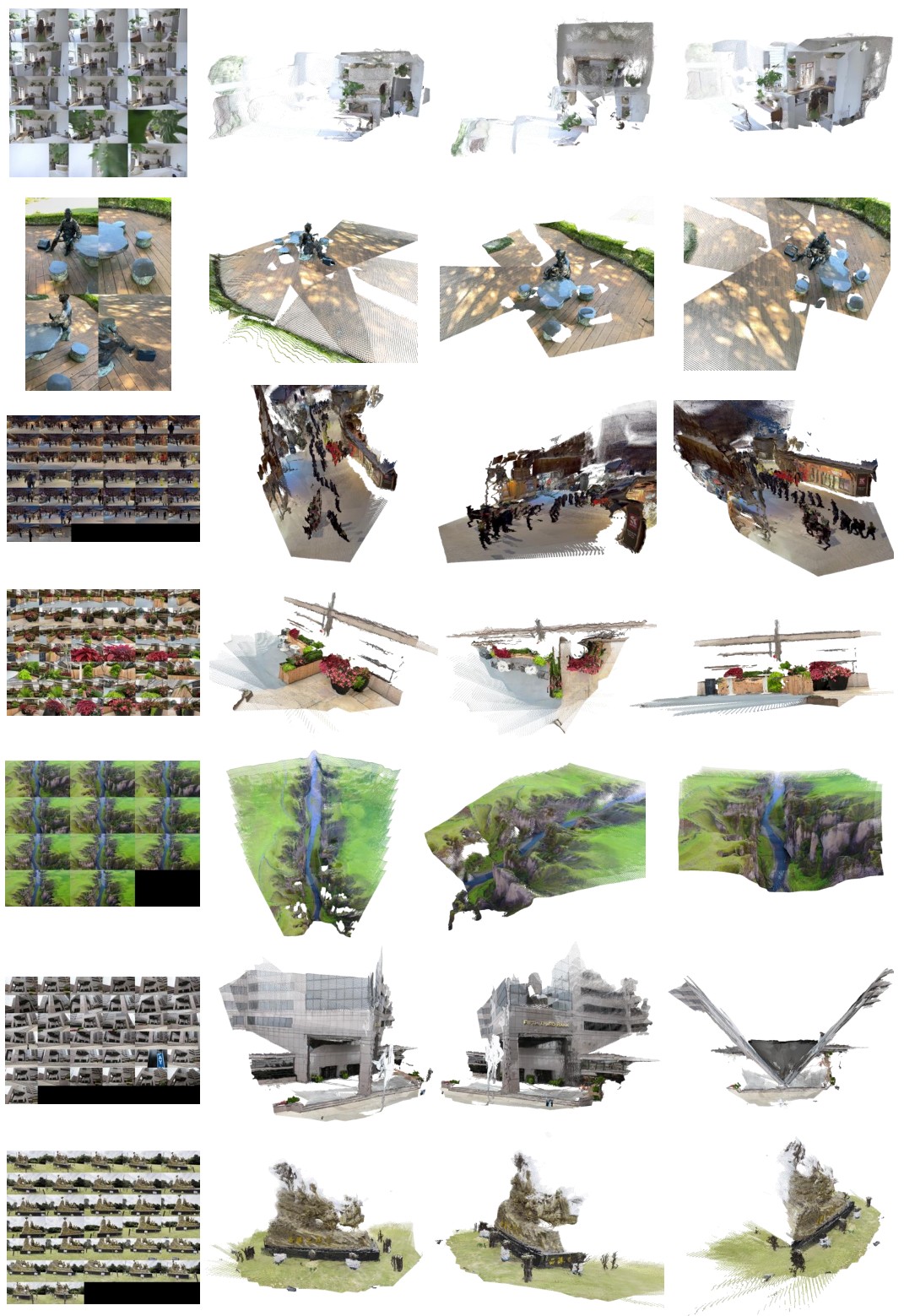

Figure 8: **More visualization results.**

