# OpenReview forum: "WinT3R: Window-Based Streaming Reconstruction with Camera Token Pool"
_ICLR.cc/2026/Conference — ICLR 2026 Poster_

### Official Review · Reviewer_6j5J · 2025-10-16

**Soundness:** 3
**Presentation:** 3
**Contribution:** 2
**Rating:** 4
**Confidence:** 5

**Summary:**

This paper proposes a feed-forward reconstruction model WinT3R, which can output precise camera poses and high-quality point maps.
The key design is a sliding window mechanism to ensure sufficient information exchange among adjacent frames, and a global camera token pool to enhance the reliability of camera pose estimation without sacrificing efficiency.
Experimental results show the effectiveness of the proposed method.

**Strengths:**

1. The presentation is clear and easy to follow.
2. The author conducted extensive experiments to support their statements. And results show that WinT3R outperforms other online baselines in some benchmarks.

**Weaknesses:**

1. The architecture is highly similar to CUT3R and VGGT. Actually, it seems like a combination of these two methods (state tokens-CUT3R, interaction within window-VGGT). And the training costs are high (compared with CUT3R's 8GPUs).
2. I want to know the relationship between the global and local tokens (image tokens and camera tokens) for each frame. From Sec.3.3, it seems that this work only supervise the point map in the local coordinate system and the relative poses between frames (following pi3). In this case, is it redundant to maintain both local and global tokens for each frame simultaneously?
3. With the window based design, I want to know the robustness of WinT3R when faced with unordered inputs. Besides, please use very long sequences (200 frames and more) to compare the reconstruction quality, as it is important in this online settings.

Please answer the above questions carefully and provide more thorough discussion and comparison.

**Questions:**

Please refer to the Weaknesses.

---

> ### Author Response · Authors · 2025-11-21
>
> **Q1:Architecture and training costs problem**
>
> To address the state tokens' historical forgetting issue in CUT3R, we innovatively propose using compact camera tokens to preserve global historical information. Our experiments demonstrate the effectiveness of this design: in long-sequence reconstruction, our model significantly outperforms CUT3R in quality and approaches the reconstruction quality of StreamVGGT. However, StreamVGGT suffers from extremely slow long-sequence reconstruction speeds, often falling below 1 FPS, which negates its real-time performance. In contrast, our model maintains its speed without significant degradation during long-sequence reconstruction. This clearly indicates that our camera token pool effectively mitigates the historical forgetting problem of CUT3R's state tokens while maintaining a highly compact representation. The results strongly validate our insight: predicting camera parameters benefits from more global information, whereas predicting local point maps relies more heavily on local information.
>
> Regarding training cost, although CUT3R uses only 8 A100 GPUs, its training lasts for one month (as indicated by the authors in a GitHub issue). Our training regimen involves 64 A800 GPUs for 7 days, plus 32 A800 GPUs for an additional 4 days. Considering the overhead of multi-node communication and the slightly lower multi-GPU performance of A800s compared to A100s, our total training cost is not substantially higher than that of CUT3R.
>
> When converted to A100 GPU hours for a standardized comparison, the training duration ranks as follows: CUT3R < Ours < VGGT < FLARE.
>
> **Q2:Relationship between the global and local tokens (image tokens and camera tokens) for each frame**
>
> In our implementation, the camera pose predicted by the model is the global camera pose for each frame, not the relative pose between different frames. However, when calculating the loss, we convert the global camera pose into pairwise relative camera poses for supervision. The camera pose represents the position and orientation of the camera for each input image within the entire 3D scene. Predicting the camera requires more global information, which is why global tokens are beneficial for camera pose prediction. Conversely, predicting the local point map is highly dependent on local information, making local tokens (image tokens) equally essential.
>
> **Q3:Robustness of WinT3R when faced with unordered input**
>
> We conducted inference experiments on unordered inputs by shuffling the entire input sequence and performed comparisons on ETH3D, DTU, and 7-Scenes. The results show that our model outperforms the currently well-performing CUT3R and StreamVGGT when dealing with unordered inputs .
>
> | Method      | ETH3D |       |         | DTU   |       |         | 7Scenes |       |         |
> |-------------|-------|-------|---------|-------|-------|---------|---------|-------|---------|
> |             | ACC$\\downarrow$   | COMP$\\downarrow$  | Overall$\\downarrow$ | ACC$\\downarrow$   | COMP$\\downarrow$  | Overall$\\downarrow$ | ACC$\\downarrow$     | COMP$\\downarrow$  | Overall$\\downarrow$ |
> | CUT3R       | 0.652 | 0.613 | 0.633   | 3.723 | 2.165 | 2.944   | $\underline{0.038}$ | 0.037 | $\underline{0.038}$ |
> | StreamVGTT  | $\\underline{0.610}$ | $\\underline{0.592}$ | $\underline{0.601}$ | $\underline{3.468}$ | **1.572** | $\underline{2.520}$ | 0.049   | **0.030**   | 0.040   |
> | Ours        | **0.504** | **0.436** | **0.470**   | **3.136** | $\underline{1.817}$ | **2.477**  | **0.033**  | $\\underline{0.031}$   | **0.032**   |

---

> ### Author Response · Authors · 2025-11-21
>
> **Q4:Very long sequences reconstruction quality**
>
> We conducted a comparison on long-sequence reconstruction, with detailed results provided in **LONG SEQUENCE COMPARISONS** of the Appendix. The results show that our model achieves nearly the same speed as CUT3R in long-sequence reconstruction, but with significantly higher quality. Compared to StreamVGGT, which caches all tokens, our approach uses a persistent state and highly compact camera tokens, avoiding the issue of an exploding number of tokens in long-sequence reconstruction as seen in StreamVGGT. Predicting the camera requires more global information, whereas predicting the local point map relies more on local information—this is the key insight behind our method. Such a computational approach is clearly more efficient.The results show that our model achieves a reconstruction speed **14 times faster** than StreamVGGT in **200-frame** reconstructions, while the peak GPU memory usage is less than **1/3** of StreamVGGT's. Furthermore. Our quality is very close to StreamVGGT, sufficiently demonstrating the effectiveness of camera token pool in long sequence reconstruction.
>
> | Frames |                   |GPU Memory|             | |      Inference Time      |        |
> |--------|----------------------|-----------|--------|----------------|-----------|--------|
> |        | CUT3R                | StreamVGGT| Ours   | CUT3R          | StreamVGGT| Ours   |
> | 50     | 35.83                | 71.08     | 24.42  | 4.81           | 17.96     | 3.10   |
> | 100    | 36.08                | 71.25     | 24.69  | 7.52           | 45.41     | 5.96   |
> | 150    | 36.32                | 71.43     | 24.94  | 10.17          | 93.31     | 9.17   |
> | 200    | 36.57                | 71.60     | 25.19  | 12.34          | 171.20    | 11.99  |
> | 250    | 36.82                | 71.77     | 25.44  | 15.74          | 265.56    | 15.47  |
> | 300    | 37.07                | 71.94     | 25.69  | 19.00          | 393.14    | 18.95  |
> | 350    | 37.31                | OOM       | 25.93  | 22.56          | OOM       | 23.06  |
> | 400    | 37.56                | OOM       | 26.18  | 25.68          | OOM       | 26.77  |
>
> 200 frame reconstruction quality comparison.
> | Model       | 7scenes |       |       | NRGBD |       |       |
> |-------------|---------|-------|-------|-------|-------|-------|
> |             | ACC$\\downarrow$     | COMP$\\downarrow$   | Overall$\\downarrow$     | ACC$\\downarrow$    | COMP$\\downarrow$   | Overall$\\downarrow$     |
> | CUT3R       | 0.083 | 0.042 | 0.062 | 0.194 | 0.089 | 0.142 |
> | StreamVGGT  | $\\underline{0.041}$   | **0.020** | **0.031** | 0.110 | **0.027** | **0.068** |
> | Ours        | **0.037**   | $\\underline{0.031}$ | $\\underline{0.034}$ | **0.095** | $\\underline{0.075}$ | $\\underline{0.085}$ |

---

### Official Review · Reviewer_K247 · 2025-10-29

**Soundness:** 3
**Presentation:** 3
**Contribution:** 3
**Rating:** 4
**Confidence:** 3

**Summary:**

This paper presents WinT3R, a feed-forward model for online 3D reconstruction from streaming images. It aims to solve the critical trade-off between reconstruction quality and real-time performance. The method introduces two key contributions: an online sliding window mechanism, which allows image tokens of adjacent frames to interact directly, improving geometric quality , and a compact camera token pool, which maintains global information from all historical frames to enhance the reliability of camera pose estimation.  Experiments demonstrate its effectiveness.

**Strengths:**

1.	This paper presents a novel framework, WinT3R, for online 3D reconstruction, which effectively tackles the critical trade-off between reconstruction quality and real-time performance. The two core contributions, the sliding window mechanism and the camera token pool, are technically sound and directly address clear limitations in prior streaming methods.
2.	Extensive experiments validate the effectiveness of the proposed method.
3.	This paper is written and organized well.

**Weaknesses:**

1.	The paper's entire motivation is to close the quality gap between "suboptimal" online methods and high-quality offline methods (like the cited VGGT or FLARE). However, all quantitative tables (Tables 1-3) only compare against other online methods. Without including at least one offline advanced baseline, it is impossible to quantify how much of the quality gap has actually been closed, which is essential for contextualizing the paper's contribution. It will be more convincing to add these comparisons.
2.	The paper states the camera head queries the entire historical pool. This implies an attention mechanism with O(T^2) computational complexity, where T is the total number of frames. This quadratic complexity is not scalable, contradicts the real-time (17.2 FPS) claim for any reasonably long video, and is a fundamental design flaw for a streaming system. It will be interesting to discuss about it.
3.	The paper states it uses the final pose from the subsequent window for overlapping frames（L209）. However, the model's token update rule adds the tokens from the initial window to the pool(Eq.5). This raises a problem that the final chosen pose does not match the token stored in the global pool. This problem undermines the pool's global consistency.

**Questions:**

Refer to the Weakness.

---

> ### Author Response · Authors · 2025-11-21
>
> **Q1:Compare against other online methods.**
>
> Thank you very much for your suggestion! In the experiments section, we have supplemented the comparison with offline models to demonstrate the performance gap between WinT3R and these offline models. The results show that our model's performance is very close to the offline model FLARE (Moreover, it achieves more than a **4x speedup** in reconstructions involving **over 50 frames**.), and its reconstruction quality is significantly better than that of Fast3R.
>
> **Q2:Quadratic complexity is not scalable, which is a fundamental design flaw for a streaming system.**
>
> In our method, we retain only one camera token per image, and the parameters of our camera head transformer are less than the backbone. To put this into perspective, consider a window containing 4 images, each with a resolution of 512x384. Assuming a model patch size of 16, the total number of image tokens within one window is calculated as follows: (512/16) * (384/16) * 4 = 3072. Even when accounting for window overlap, which effectively halves the input, the model would need to process 3072 / 2 = **1536** images to generate a number of camera tokens equivalent to the number of image tokens in a single window. This volume of image input is entirely sufficient for an online model, so concerns regarding real-time performance are unnecessary.
> We conducted a comparison on long-sequence reconstruction, with detailed results provided in **LONG SEQUENCE COMPARISONS** of the Appendix. The results show that our model achieves nearly the same speed as CUT3R in long-sequence reconstruction, but with significantly higher quality. Compared to StreamVGGT, which caches all tokens, our approach uses a persistent state and highly compact camera tokens, avoiding the issue of an exploding number of tokens in long-sequence reconstruction as seen in StreamVGGT. Predicting the camera requires more global information, whereas predicting the local point map relies more on local information—this is the key insight behind our method. Such a computational approach is clearly more efficient.The results show that our model achieves a reconstruction speed **14 times faster** than StreamVGGT in **200-frame** reconstructions, while the peak GPU memory usage is less than **1/3** of StreamVGGT's. Furthermore. Our quality is very close to StreamVGGT, sufficiently demonstrating the effectiveness of camera token pool in long sequence reconstruction.
>
> | Frames |                   |GPU Memory|             | |      Inference Time      |        |
> |--------|----------------------|-----------|--------|----------------|-----------|--------|
> |        | CUT3R                | StreamVGGT| Ours   | CUT3R          | StreamVGGT| Ours   |
> | 50     | 35.83                | 71.08     | 24.42  | 4.81           | 17.96     | 3.10   |
> | 100    | 36.08                | 71.25     | 24.69  | 7.52           | 45.41     | 5.96   |
> | 150    | 36.32                | 71.43     | 24.94  | 10.17          | 93.31     | 9.17   |
> | 200    | 36.57                | 71.60     | 25.19  | 12.34          | 171.20    | 11.99  |
> | 250    | 36.82                | 71.77     | 25.44  | 15.74          | 265.56    | 15.47  |
> | 300    | 37.07                | 71.94     | 25.69  | 19.00          | 393.14    | 18.95  |
> | 350    | 37.31                | OOM       | 25.93  | 22.56          | OOM       | 23.06  |
> | 400    | 37.56                | OOM       | 26.18  | 25.68          | OOM       | 26.77  |
>
> 200 frame reconstruction quality comparison.
> | Model       | 7scenes |       |       | NRGBD |       |       |
> |-------------|---------|-------|-------|-------|-------|-------|
> |             | ACC$\\downarrow$     | COMP$\\downarrow$   | Overall$\\downarrow$     | ACC$\\downarrow$    | COMP$\\downarrow$   | Overall$\\downarrow$     |
> | CUT3R       | 0.083 | 0.042 | 0.062 | 0.194 | 0.089 | 0.142 |
> | StreamVGGT  | $\\underline{0.041}$   | **0.020** | **0.031** | 0.110 | **0.027** | **0.068** |
> | Ours        | **0.037**   | $\\underline{0.031}$ | $\\underline{0.034}$ | **0.095** | $\\underline{0.075}$ | $\\underline{0.085}$ |

---

> ### Author Response · Authors · 2025-11-21
>
> **Q3:Camera token pool's global consistency problem.**
>
> We conducted an experiment on pose selection, and the results demonstrate that the final selected pose consistently yields better outcomes. In the camera token pool, the overlapping section contains two distinct camera tokens for the same frame, each originating from the output of different windows. These tokens primarily encode the relative positional information between their respective windows, and they complement each other, thereby enriching the overall information contained within the camera tokens. Furthermore, since the camera head employs a Transformer architecture, the attention mechanism inherently averages the information across different tokens to some extent. Consequently, the final pose prediction is based on a more comprehensive set of observations compared to the initial pose, and thus there is no need to worry about the global consistency of the pool.
>
> |              | 7scenes |       | NRGBD |       | DTU   |       | ETH3D |       |
> |--------------|---------|-------|-------|-------|-------|-------|-------|-------|
> |              | ACC$\\downarrow$     | COMP$\\downarrow$  | ACC$\\downarrow$   | COMP$\\downarrow$  | ACC$\\downarrow$   | COMP$\\downarrow$  | ACC$\\downarrow$   | COMP$\\downarrow$  |
> | **Former Pose** | 0.027   | 0.025 | 0.039 | 0.022 | 3.984 | 1.890 | 0.531 | 0.349 |
> | **Later Pose**  | **0.023** | **0.022** | **0.032** | **0.020** | **3.638** | **1.838** | **0.411** | **0.272** |

---

### Official Review · Reviewer_i3GJ · 2025-10-31

**Soundness:** 2
**Presentation:** 4
**Contribution:** 3
**Rating:** 6
**Confidence:** 4

**Summary:**

This paper proposes a new online streaming 3D reconstruction method called WinT3R that combines a sliding-window attention mechanism with a camera token pool memory to achieve real-time performance. It is an improvement based on the prior work CUT3R. The novelty mainly comes from the attention between image tokens within windows, and representing prior frames via compact tokens in a global camera token pool. The experimental results on standard benchmarks show strong improvements over existing online methods.

**Strengths:**

It provides a way of balancing the usually high-cost transformer-based 3D reconstruction model and reconstruction quality and consistency over long sequences. This enables the model to operate at real-time with streaming input. The sliding window attention effectively avoids the full casual attention cost. It maintains the simplicity of feedforward neural reconstruction models without heavy keypoint feature matching in traditional SLAM pipelines.

**Weaknesses:**

* It is unclear for long sequences how well the camera token pool can handle and preserve the global geometry for frames that are far apart.
* For the efficiency and scalability claim, It lacks some experiments on demonstrating the ability of the model on long sequences and the corresponding runtime statistics.
* The experiments on the selected scenes are mostly all indoor scenes (e.g. 7 Scenes, NRGBD). It lacks some experiments on more varying conditions such as outdoor scenes.

**Questions:**

It seems like the selection of the window's length and stride is important here. Different video sequences will typically have non-uniform motions at every point. Will this affect the reconstruction quality a lot?

---

> ### Author Response · Authors · 2025-11-21
>
> **Q1:Window's length and stride selection problem and non-uniform motions of different video sequences**
>
> In our ablation study, we compared the performance without window (i.e., window size = 1) and with a window size of 4. The results show that the performance with a window size of 4 is significantly better. Regarding the issue of window length, we also conducted an additional set of experiments with different window sizes to investigate the impact on the reconstruction results. In the comparison of window sizes 2, 4, and 8, the results indicate that the window size has little impact on the accuracy metric, but it has a significant impact on completeness. While a larger window size can lead to better reconstruction quality, it also introduces significant reconstruction latency. This is because the first input frame within a window must wait for subsequent inputs to form the complete window before the entire set can be fed into the model. The resulting delay increases almost linearly with the window size. Since we aim for our model to be a low-latency online model, this trade-off is a critical consideration. As a result, we selected a window size of 4. In practical applications, an appropriate window size can be selected based on the requirements for latency and reconstruction quality.
>
> |              | 7scenes |       | NRGBD |       | DTU   |       | ETH3D |       | latency(ms) |
> |--------------|---------|-------|-------|-------|-------|-------|-------|-------|-------------|
> |              | ACC$\downarrow$     | COMP$\downarrow$  | ACC$\downarrow$   | COMP$\downarrow$  | ACC$\downarrow$   | COMP$\downarrow$  | ACC$\downarrow$   | COMP$\downarrow$  |             |
> | window_size2 | 0.120   | 0.415 | $\underline{0.196}$ | 0.453 | **11.80** | 26.64 | **0.962** | $\underline{1.633}$ | 76          |
> | window_size4 | $\underline{0.112}$ | $\underline{0.392}$ | 0.224 | $\underline{0.356}$ | $\\underline{12.35}$ | $\underline{23.13}$ | $\underline{0.977}$ | 1.716 | 121         |
> | window_size8 | **0.111** | **0.209** | **0.176** | **0.325** | 12.90 | **15.08** | 0.989 | **1.506** | 226         |
>
> Regarding the fact that different video sequences have non-uniform motions at every point, we strive to ensure diversity in camera motion within our training data frame selection. This includes videos with uniform motion as well as multiview images exhibiting non-uniform motions. During the final testing of our model, as long as a certain degree of overlap (which can be relatively small) can be mutually found among the images within a window, the reconstruction quality remains unaffected.
>
> **Q2:It is unclear for long sequences how well the camera token pool can handle and preserve the global geometry for frames that are far apart.**
>
> We conducted a comparison on long-sequence reconstruction, with detailed results provided in **LONG SEQUENCE COMPARISONS** of the Appendix. The results show that our model achieves nearly the same speed as CUT3R in long-sequence reconstruction, but with significantly higher quality. Compared to StreamVGGT, which caches all tokens, our approach uses a persistent state and highly compact camera tokens, avoiding the issue of an exploding number of tokens in long-sequence reconstruction as seen in StreamVGGT. Predicting the camera requires more global information, whereas predicting the local pointmap relies more on local information—this is the key insight behind our method. Such a computational approach is clearly more efficient.The results show that our model achieves a reconstruction speed **14 times faster** than StreamVGGT in **200-frame** reconstructions, while the peak GPU memory usage is less than **1/3** of StreamVGGT's. Furthermore. Our quality is very close to StreamVGGT, sufficiently demonstrating the effectiveness of camera token pool in long sequence reconstruction.
>
> 200 frame reconstruction quality comparison.
>
> | Model       | 7scenes |       |       | NRGBD |       |       |
> |-------------|---------|-------|-------|-------|-------|-------|
> |             | ACC$\downarrow$     | COMP$\downarrow$  | Overall$\downarrow$    | ACC$\downarrow$   | COMP$\downarrow$  | Overall$\downarrow$    |
> | CUT3R       | 0.083 | 0.042 | 0.062 | 0.194 | 0.089 | 0.142 |
> | StreamVGGT  | $\underline{0.041}$   | **0.020** | **0.031** | $\underline{0.110}$ | **0.027** | **0.068** |
> | Ours        | **0.037**   | $\underline{0.031}$ | $\underline{0.034}$ | **0.095** | $\underline{0.075}$ | $\underline{0.085}$ |

---

> ### Author Response · Authors · 2025-11-21
>
> **Q3:Lacks some experiments on demonstrating the ability of the model on long sequences and the corresponding runtime statistics.**
>
> We present the model's capability on handling long sequences and the corresponding runtime statistics. Detailed results and visualizations are provided in **LONG SEQUENCE COMPARISONS** of the Appendix, which demonstrate that our model remains efficient in reconstructions spanning hundreds of frames and achieves significantly higher quality than CUT3R.
>
> | Frames |                   |GPU Memory|             | |      Inference Time      |        |
> |--------|----------------------|-----------|--------|----------------|-----------|--------|
> |        | CUT3R                | StreamVGGT| Ours   | CUT3R          | StreamVGGT| Ours   |
> | 50     | 35.83                | 71.08     | 24.42  | 4.81           | 17.96     | 3.10   |
> | 100    | 36.08                | 71.25     | 24.69  | 7.52           | 45.41     | 5.96   |
> | 150    | 36.32                | 71.43     | 24.94  | 10.17          | 93.31     | 9.17   |
> | 200    | 36.57                | 71.60     | 25.19  | 12.34          | 171.20    | 11.99  |
> | 250    | 36.82                | 71.77     | 25.44  | 15.74          | 265.56    | 15.47  |
> | 300    | 37.07                | 71.94     | 25.69  | 19.00          | 393.14    | 18.95  |
> | 350    | 37.31                | OOM       | 25.93  | 22.56          | OOM       | 23.06  |
> | 400    | 37.56                | OOM       | 26.18  | 25.68          | OOM       | 26.77  |
>
> **Q4:Lacks some experiments on more varying conditions such as outdoor scenes.**
>
> Our evaluation data also includes a substantial number of outdoor scenes. For instance, approximately half of the scenes in the ETH3D and Tanks and Temples datasets are outdoors. Similarly, many categories within the CO3Dv2 dataset correspond to outdoor scenes or indoor-outdoor hybrid environments
>
> We have removed the indoor scenes from the test data of ETH3D and Tanks and Temples, retaining only the outdoor scenes for evaluation. The results are as follows:
>
> | Method      | ETH3D  |        |         | Tanks and Temples |         |         |
> |-------------|--------|--------|---------|-------------------|---------|---------|
> |             | ACC$\downarrow$   | COMP$\downarrow$   | Overall$\downarrow$ | RRA@30$\uparrow$ | RTA@30$\uparrow$ | AUC@30$\uparrow$ |
> | CUT3R       | $\underline{0.351}$  | 0.294  | $\underline{0.322}$   | 90.46            | 91.57   | 77.41   |
> | Stream VGIT | 0.407  | $\underline{0.267}$  | 0.337   | **99.80** | **97.58** | $\underline{80.64}$ |
> | Ours        | **0.175** | **0.097** | **0.136** | $\underline{93.59}$           | $\underline{93.89}$   | **81.85** |
>
> The results show that our model remains better or comparable against other online models in outdoor scenes .

---

### Official Review · Reviewer_Qrbo · 2025-11-01

**Soundness:** 3
**Presentation:** 2
**Contribution:** 2
**Rating:** 6
**Confidence:** 3

**Summary:**

This paper (“WinT3R”) presents a feed-forward architecture for predicting point maps from streaming RGB video. At a high level, it combines sliding-window attention with a global persistent state, conceptually related to CUT3R and the streaming variant of VGGT (StreamVGGT). The method uses overlapping windows to encourage direct interaction among neighboring frames while maintaining memory efficiency. In addition, a global camera token pool stores compact historical camera tokens to support future camera pose prediction. The overall idea is simple and well-motivated.

Overall, the design choices are reasonable and the empirical results are strong. My concerns are primarily about figure presentation and some missing ablations; details follow.

**Strengths:**

- The approach is simple, straightforward, and reasonable: combining sliding-window attention with a global state is a natural way to balance local interaction and long-range consistency.

- Experimental results indicate state-of-the-art reconstruction quality among online/streaming methods.

- The global camera token pool, while somewhat pragmatic, is an effective design specifically for camera pose estimation in a streaming setting.

**Weaknesses:**

- Result visualization could be improved. For example, in Figure 4 it is difficult to tell which columns are better without clearer legends or more distinct visual cues.

- Consider adding videos or point-cloud visualizations in the supplementary material to better convey qualitative improvements.

- The performance improvement over StreamVGGT is relatively modest, which could lessen the perceived contribution of this paper.

**Questions:**

- What are the training time and GPU resources used in the ablation study?

- Which model size (and window size) is used for those ablations?

- There is no ablation that removes the global persistent state entirely. How would performance change without it (e.g., sliding-window only, no global state)?

---

> ### Author Response · Authors · 2025-11-21
>
> **Q1: What are the training time and GPU resources used in the ablation study?**
>
> In the ablation study, all four models have the same number of parameters. Regarding training time, the model without the camera token pool took only **3 hours (1.76%)** less to train than the full model, further demonstrating the compact design of our camera token. The model that did not utilize the window-based approach required a longer training time because predicting frame-by-frame necessitates more passes through the model compared to the window-based method. This also indicates that the window-based computation is more efficient.
>
> | Model       | Training time | size    |
> |-------------|---------------|---------|
> | w/o pool    | 167h          | 749.46M |
> | w/o window  | 192h          | 749.46M |
> | w/o overlap | 117h          | 749.46M |
> | full model  | 170h          | 749.46M |
>
> **Q2:Which model size (and window size) is used for those ablations?**
>
> In our ablation study, all models have the same size (749.46M parameters) and differ only in their forward pass configurations. The configurations for ablation study are listed below:
> - w/o pool: Window size = 4, but the camera token pool is removed.
> - w/o window: The window-based design is removed, which can be interpreted as a window size of 1.
> - w/o overlap: Window size = 4, stride = 4.
> - Full model: Window size = 4, stride = 2.
>
> **Q3:There is no ablation that removes the global persistent state entirely.**
>
> The global persistent state follows the approach used in CUT3R and is designed to maintain the global information of the scene. Since prior work has already validated the effectiveness of this design, we did not conduct an additional ablation study. Functionally, the Global Persistent State connects adjacent windows. When the first window is input into the model, the Global Persistent State consists of the model's learnable tokens. For each subsequent window processed by the model, the Global Persistent State interacts with the current window's image tokens and camera tokens via cross-attention. Consequently, the Global Persistent State continuously evolves during online prediction, and its removal would disrupt the connection of image tokens across adjacent windows. From an architectural perspective, the Global Persistent State serves as the complete input for one branch (the dual branch) of the model's decoder. Removing the Global Persistent State would thus render the model structurally incomplete. Therefore, the Global Persistent State is an indispensable component and cannot be omitted. The camera tokens act as a constant memory, mitigating the potential forgetting issue caused by the continually changing Global Persistent State. In this way, the camera tokens and the Global Persistent State complement each other.
>
> **Q4:Result visualization could be improved.**
>
> Thank you for your suggestion. We have revised Fig. 4 to better demonstrate the superior performance of our model.
>
> **Q5:Consider adding videos or point-cloud visualizations.**
>
> Thank you for your suggestion. We provide additional visualization results to better demonstrate the superior performance of our model in appendix.

---

> ### Author Response · Authors · 2025-11-21
>
> **Q6:The performance improvement over StreamVGGT is relatively modest.**
>
> First of all, our model size of **750M** is smaller than StreamVGGT's **1.2B**. Moreover, our model requires significantly less computation in long sequences compared to StreamVGGT. In reconstructions involving hundreds of frames, WinT3R remains stable, with peak GPU memory usage around **26GB**, whereas StreamVGGT's peak GPU memory usage exceeds **71GB**, making it highly prone to out-of-memory (OOM) errors and extremely slow in reconstruction speed. This is because StreamVGGT adopts an approach that caches all tokens, whereas our method utilizes a persistent state and highly compact camera tokens, avoiding the explosion in the number of tokens that StreamVGGT experiences in long-sequence reconstructions. Predicting camera poses requires more global information, while predicting local point maps relies more on local information. This is the key insight behind our approach, and such a computational strategy is evidently more efficient. We conducted an additional inference experiment to substantiate this claim, with detailed results and visualizations provided in **LONG SEQUENCE COMPARISONS** of the Appendix. The results show that our model achieves a reconstruction speed **14 times faster** than StreamVGGT in **200-frame** reconstructions, while the peak GPU memory usage is less than **1/3** of StreamVGGT's. Furthermore, in reconstructions involving dozens of frames, our quality surpasses StreamVGGT, and in reconstructions with hundreds of frames, our quality is very close to StreamVGGT, sufficiently demonstrating that our model is superior to StreamVGGT.
>
> | Frames |                   |GPU Memory|             | |      Inference Time      |        |
> |--------|----------------------|-----------|--------|----------------|-----------|--------|
> |        | CUT3R                | StreamVGGT| Ours   | CUT3R          | StreamVGGT| Ours   |
> | 50     | 35.83                | 71.08     | 24.42  | 4.81           | 17.96     | 3.10   |
> | 100    | 36.08                | 71.25     | 24.69  | 7.52           | 45.41     | 5.96   |
> | 150    | 36.32                | 71.43     | 24.94  | 10.17          | 93.31     | 9.17   |
> | 200    | 36.57                | 71.60     | 25.19  | 12.34          | 171.20    | 11.99  |
> | 250    | 36.82                | 71.77     | 25.44  | 15.74          | 265.56    | 15.47  |
> | 300    | 37.07                | 71.94     | 25.69  | 19.00          | 393.14    | 18.95  |
> | 350    | 37.31                | OOM       | 25.93  | 22.56          | OOM       | 23.06  |
> | 400    | 37.56                | OOM       | 26.18  | 25.68          | OOM       | 26.77  |
>
> 200 frame reconstruction quality comparison.
> | Model       | 7scenes |       |       | NRGBD |       |       |
> |-------------|---------|-------|-------|-------|-------|-------|
> |             | ACC$\\downarrow$     | COMP$\\downarrow$   | Overall$\\downarrow$     | ACC$\\downarrow$    | COMP$\\downarrow$   | Overall$\\downarrow$     |
> | CUT3R       | 0.083 | 0.042 | 0.062 | 0.194 | 0.089 | 0.142 |
> | StreamVGGT  | $\\underline{0.041}$   | **0.020** | **0.031** | 0.110 | **0.027** | **0.068** |
> | Ours        | **0.037**   | $\\underline{0.031}$ | $\\underline{0.034}$ | **0.095** | $\\underline{0.075}$ | $\\underline{0.085}$ |

---

### Author Response · Authors · 2025-11-21

We sincerely thank the reviewers for their time and effort in reviewing our paper and making valuable suggestions.  We also appreciate your positive feedback on our results, particularly your recognition of novelty(K247), technical soundness(Qrbo, i3GJ), and strong experimental performance(Qrbo, K247, 6j5J).

Regarding the reviewers' concerns about the model's long sequence performance, we have reported detailed experimental results and visualizations in the appendix. Additionally, we have included performance comparisons between our model and mainstream offline models.

We have uploaded a revised draft of our paper, with changes marked in blue. We summarize the revision here:

- added offline models comparison in experiments;
- added inference time and peak GPU memory usage statistics for long sequences in the appendix;
- added reconstruction quality statistics for long sequences in the appendix;
- added visualization results in the appendix;

We sincerely value your reviews and believe they have greatly contributed to making the new revision more comprehensive. Additionally, we have responded to your individual comments and questions by commenting on your reviews.

---

### Meta-Review · Area_Chair_CK89 · 2025-12-29

**Summary:**

The paper received mixed initial reviews, with scores of 6, 6, 4, and 4. Overall, multiple reviewers noted that the paper is well motivated and achieves a good balance between reconstruction quality and computational cost; all reviewers agreed that the method demonstrates strong performance compared to existing online baselines. At the same time, reviewers raised several major concerns, including limited evidence on long-sequence behavior and scalability, insufficient comparisons to offline methods, questions about novelty and similarity to prior work such as CUT3R and StreamVGGT, as well as the lack of more comprehensive visualizations.

In the rebuttal and revised submission, the authors provided extensive additional experiments and clarifications. These include critical long-sequence evaluations (up to hundreds of frames) with detailed runtime and peak memory statistics, new comparisons against representative offline methods, additional ablations on window size and design choices, and clearer explanations of the role and consistency of camera tokens, among other improvements. In the AC’s assessment, the rebuttal addresses most of the major technical concerns raised by the reviewers, particularly those related to scalability, efficiency, and empirical validation. As a result, the AC anticipates that the final scores are likely to increase, for example to 8, 6, 6, 6, or possibly 8, 6, 6, 4 (see comments below for detailed analysis in Reviewer Concerns and Reviewer Scores).

Overall, while there remains a possibility that the final reviews are still somewhat mixed (e.g., with one reviewer maintaining a score of 4), the AC appreciates the overall strength of the method, the thoroughness of the rebuttal, and the solid performance demonstrated both before and after revision. Therefore, the AC recommends acceptance of this paper. In general, the AC agrees with reviewers Qrbo and i3GJ that the paper provides a simple yet effective approach to balancing cost and quality in long-sequence online reconstruction using transformer-based architectures.

**Reviewer Concerns:**

### Reviewer Qrbo (Score: 6)

- The reviewer raised concerns about visualization clarity, missing qualitative results, modest performance gains over StreamVGGT, and the lack of certain ablations, including training cost, window size, and removal of the global persistent state.
- In the rebuttal, the authors improved visualizations, added additional qualitative results in the appendix, provided detailed ablation tables and training-cost breakdowns, and explained why removing the global persistent state is not structurally feasible. The authors also added long-sequence experiments with detailed reconstruction quality, memory usage, and inference-time comparisons, emphasizing efficiency advantages over StreamVGGT.

---

### Reviewer i3GJ (Score: 6)

- The reviewer expressed concerns about unclear behavior on long sequences, lack of runtime statistics to support scalability claims, limited diversity of evaluation scenes, and sensitivity to window size and stride.
- In response, the authors added comprehensive long-sequence experiments with detailed memory and inference-time statistics, provided outdoor-only evaluations, and included window-size ablations together with a discussion of latency–quality trade-offs.

---

### Reviewer K247 (Score: 4)

- The reviewer raised concerns regarding the absence of offline baselines, potential quadratic complexity of the camera token pool, and possible global consistency issues in the camera token pool.
- In the rebuttal, the authors added comparisons against representative offline methods, addressed the complexity concern through token-count analysis and long-sequence experiments, and discussed the global consistency issue with additional pose-selection experiments.

---

### Reviewer 6j5J (Score: 4)

- The reviewer expressed concerns about perceived incremental novelty relative to CUT3R and VGGT, high training cost, an unclear relationship between global and local tokens, robustness to unordered inputs, and the lack of very long-sequence quality evaluations.
- In the rebuttal, the authors clarified the distinct roles of camera tokens and image tokens, provided normalized training-cost comparisons, added experiments on unordered inputs, and included extensive long-sequence evaluations covering both reconstruction quality and efficiency.

---

Overall, the AC finds that the authors provided a thorough and responsive rebuttal, addressing nearly all of the concerns raised by the reviewers with additional experiments, analyses, and clarifications.

**Reviewer Scores:**

### Reviewer Qrbo

- **Original score:** 6
- **Predicted final score:** 6
- **Rationale:** The rebuttal addresses most of the reviewer’s technical concerns through additional ablations, improved visualizations, and extensive long-sequence experiments, particularly strengthening the efficiency and scalability claims. However, the authors did not provide video results, which may have been of particular interest to this reviewer. In addition, while the efficiency advantages are clearly reinforced, the concern that the quality improvement over StreamVGGT is relatively modest may not be fully addressed. As a result, the AC expects the reviewer to remain positive but not necessarily to increase their score.

---

### Reviewer i3GJ

- **Original score:** 6
- **Predicted final score:** 8
- **Rationale:** The reviewer’s primary concerns are all directly addressed through substantial new experiments and analyses. The added evidence significantly strengthens the empirical validation of the method, making a score increase likely.

---

### Reviewer K247

- **Original score:** 4
- **Predicted final score:** 6
- **Rationale:** The rebuttal reasonably addresses the reviewer’s concerns by adding offline baselines, clarifying the computational complexity of the camera token pool, and providing empirical evidence supporting the global consistency of the design. Most of the reviewer’s technical objections appear to be addressed, making a score increase likely. The concern regarding pose selection and consistency is an interesting and valid question; the AC finds that the newly added experiments and analyses can justify the design choice.

---

### Reviewer 6j5J

- **Original score:** 4
- **Predicted final score:** 4–6
- **Rationale:** The rebuttal resolves most empirical and technical concerns through additional experiments on long sequences, robustness to unordered inputs, and normalized training-cost comparisons. However, the reviewer’s skepticism regarding the degree of novelty relative to prior work may persist, making a score increase possible but less certain.

---

### Decision · Program_Chairs · 2026-01-26

Accept (Poster)